# Attention Aware Deep Learning Approaches for an Efficient Stress Classification Model

**DOI:** 10.3390/brainsci13070994

**Published:** 2023-06-25

**Authors:** Muhammad Zulqarnain, Habib Shah, Rozaida Ghazali, Omar Alqahtani, Rubab Sheikh, Muhammad Asadullah

**Affiliations:** 1Faculty of Computing, The Islamia University of Bahawalpur, Punjab, Pakistan; rubabsheikh77@outlook.com (R.S.); muhammad.asadullah@iub.edu.pk (M.A.); 2Department and College of Computer Science, King Khalid University, Abha 62529, Saudi Arabia; habibshah.uthm@gmail.com (H.S.); osalqahtani@kku.edu.sa (O.A.); 3Faculty of Computer Science and Information Technology, Universiti Tun Hussein Onn Malaysia, Batu Pahat 86400, Johor, Malaysia; rozaida@uthm.edu.my

**Keywords:** deep learning, long short-term memory, KNHANEs-VI, stress classification

## Abstract

In today’s world, stress is a major factor for various diseases in modern societies which affects the day-to-day activities of human beings. The measurement of stress is a contributing factor for governments and societies that impacts the quality of daily lives. The strategy of stress monitoring systems requires an accurate stress classification technique which is identified via the reactions of the body to regulate itself to changes within the environment through mental and emotional responses. Therefore, this research proposed a novel deep learning approach for the stress classification system. In this paper, we presented an Enhanced Long Short-Term Memory(E-LSTM) based on the feature attention mechanism that focuses on determining and categorizing the stress polarity using sequential modeling and word-feature seizing. The proposed approach integrates pre-feature attention in E-LSTM to identify the complicated relationship and extract the keywords through an attention layer for stress classification. This research has been evaluated using a selected dataset accessed from the sixth Korea National Health and Nutrition Examination Survey conducted from 2013 to 2015 (KNHANES VI) to analyze health-related stress data. Statistical performance of the developed approach was analyzed based on the nine features of stress detection, and we compared the effectiveness of the developed approach with other different stress classification approaches. The experimental results shown that the developed approach obtained accuracy, precision, recall and a F1-score of 75.54%, 74.26%, 72.99% and 74.58%, respectively. The feature attention mechanism-based E-LSTM approach demonstrated superior performance in stress detection classification when compared to other classification methods including naïve Bayesian, SVM, deep belief network, and standard LSTM. The results of this study demonstrated the efficiency of the proposed approach in accurately classifying stress detection, particularly in stress monitoring systems where it is expected to be effective for stress prediction.

## 1. Introduction

In the last decades, many researchers have observed that there are inextricable links between the mental health of the individual and his/her physical condition [1]. Recently, it has become an important and embedded part of our professional life, especially in a severely competitive economy. Stress mostly occurs due to increased work pressure which might occur in any form [2]. There are several analyses have been conducted by different researchers to find the key source of increased stress levels [3]. In the workplace, an individual must constantly face different situations, including but not limited to job insecurity, work overload, lack of job satisfaction, and the pressure to stay up-to-date. Stress is a critical factor in modern societies. It is increasing issue and it has become an unavoidable part of our daily lives. The continued presence of stress can lead to some bad health effects, such as susceptibility to infections, high blood pressure, lack of sleep and cardiovascular disease. All of these conditions result in mental stress which has become a leading cause of several diseases. In addition, stress can be examined as s physiological and psychological response of the body to adverse environmental situations. Consequently, the effective and accurate detection of stress can lead to specific prevention and intervention strategies in personal healthcare [4,5]. Thus, it is crucial to identify and control stress at an early stage to prevent stress-related illnesses. Selye [6] introduced the terms “eustress” and “distress” to differentiate between positive and negative stress, respectively. Eustress arises from positive changes or demands that do not pose a problem for coping or adapting to new situations. It can assist us in achieving our goals and enhancing productivity [7]. 

Stress detection is generally evaluated subjectively and measured by surveys; so far, numerous studies have been conducted on the association between physical activity and lifestyle, although the majority have been constrained to certain categories and factors. [8]. Figure 1 presents the conceptual design of our research study on different aspects of stress. In this research, the significance of sleep, number of working hours, physical activity and heart rate with regard to stress levels are analyzed. Episodic stress occurs when stressful situations happen more commonly but intermittently. It is associated with a highly stressful and disorganized life [9]. Lastly, chronic stress, which is the most damaging, occurs when stressors are persistent and long-lasting, such as family issues, job strain, or poverty [7]. To prevent stress from reaching its highest level and to reduce associated risks [10], it is essential to identify and address it during its early phases, specifically when it is still in the form of acute or episodic stress.

In Europe, stress ranks as the second most prevalent lifestyle health issue, trailing behind musculoskeletal disorders which can sometimes manifest as stress symptoms [7,11]. The financial impact of lifestyle stress was significant in 2002, costing EU enterprises €20 billion [12]. Additionally, a considerable 22% of European workers experienced work-related stress in 2005 [13]. A recent survey indicates that 51% of European workers acknowledge the prevalence of stress in their workplaces, and work-related stress and psychosocial risks account for approximately 50–60% of all lost working days in European companies [11]. Similarly, South Korea is also recognized as a country with a high incidence of heart disease, ranking second in terms of total deaths [14]. Common risk aspects for developing coronary heart disease (CHD) include unhealthy habits such as physical inactivity, a poor diet, drinking, smoking, and stress [15,16].

Therefore, the majority of people have used different variables to develop an effective technique for evaluating an unspecified kind of stress. Stress classification and prediction approaches using machine learning techniques, such as naïve Bayes (NB) and support vector machine (SVM), have been studied to enhance their prediction or classification results [17,18]. Jawharali et al. [17] introduced a Fuzzy support vector machine to measure perceived stress through EOG (ElectroOculoGraphy) technique, which can accurately predict the stress level. This technique illustrates evaluation accuracy more accurately than others conventional techniques without EOG. Similarly, Sani et al. [18] suggested an approach for classifying stress patients using SVM along with redial kernel function to achieve 83.29% classification accuracy based on electroencephalography signal.

The utilization of medical IT in conjunction with machine learning technologies has significantly enhanced the accuracy of disease prediction. This is achieved through the creation of predictive models using disease-related learning data have played a crucial role in this advancement [19]. However, due to the complexity of the data being analyzed, the application of deep learning techniques becomes necessary [20,21]. Several studies have been applied in the field of cardiovascular disease, employing machine learning methods. For instance, Khateb and Montezer [22] proposed a heart disease risk prediction approach by employing the Dempster-Shafer evidence concept and manipulating a fuzzy evidential hybrid inference mechanism. Another study by Krishnaieh et al. [23] introduced a cardiovascular risk assessment model by utilizing fuzzy K-nearest neighbor (K-NN) classifiers to handle uncertainty associated with measured values. Nevertheless, there is a noticeable research gap when it comes to a prediction technique for domestic cardiovascular disease [24,25]. In recent times, there has been an increasing amount of attention paid to constructing prediction models based on big data and leveraging advancements in deep learning technology.

Moreover, Bobade et al. [26] referred to the various deep learning and machine learning approaches for stress monitoring systems on individuals utilizing multimodal datasets acquired from physiological and motion sensors, which can avoid detecting a person from different stress associated health issues. However, these approaches need complicated and stochastic processing of physiological signals, which is unsuitable for the development of big data prediction techniques and deep learning technologies.

Recently, stress classification approaches have been aided by artificial intelligence, and numerous statistical and machine learning techniques have been developed for stress data analysis in healthcare [27]. The Long Short-Term Memory (LSTM) model is a powerful learning model in deep learning methods, with advanced technology and excellent performance on sequential data [28]. It consists of a gated mechanism which controls the flow of information and employs supervised learning via the backpropagation algorithm. The LSTM is employed in several medical areas and is extensively applied in medical research because of its superior performance [29,30]. 

In this study, we investigated the useful functionality of the conventional LSTM for stress classification using stress-correlated physical activity and lifestyle data accessed from the 2013–2015 Korea National Health and Nutrition Examination Survey (KNHANES VI) database [31]. First, we studied whether the stress assessment was feasible by analyzing the stress-related physical activity and lifestyle data and dividing the persons between the ages of 18 to 75 years old into two groups: those who were frequently felt stressed and those who did not. Second, we investigated an E-LSTM-based method for classifying stress that was designed to incorporate features from physical activity and lifestyle data, which were deemed essential for accurate stress evaluation. Furthermore, in order to select the most informative features, our developed framework demonstrated further enhancement in the conventional LSTM design based on the feature-attention mechanism, namely Enhanced LSTM (E-LSTM) for stress classification. The primary goal of our study is to enhance the conventional LSTM structure in order to increase stress classification accuracy and reduce the information loss.

## 2. The Impact of Stress on Society and the Economy

Stress expresses the reactions of the human body and is increasing day by day in advanced societies [32]. Office stress is caused by a mismatch between job expectations and skills, as well as time pressure and heavy workloads. Family interrelated conflicts, disabilities, chronic injuries and mental difficulties are all examples of off-the-job stress factors. In Europe, stress is the second most serious work-related health issue [33,34]. According to the American Institute of Stress, the United States spends $300 billion annually on stress-related diseases, http://www.who.int/occupational_health/topics/stressatwp/en/ (accessed on 29 April 2020). EU companies spent 25 billion euros on work-related stress in 2013 [35]. According to a recent public survey [36], 51 percent of European employees are stressed at their respective workplaces. In the European business sector, it has been estimated that 50–60% of all missing workdays are due to work-related stress and psychosocial hazards [34].

A study conducted in 2015 utilized data from the National Health Interview Survey to assess various risk factors in the US population [37]. The analysis involved 28,993 data points, which were subjected to cluster analysis in order to examine the risk factors. In order to maintain accuracy, any data with missing values were excluded from the analysis. In 2016, Liu et al., conducted a study using the 2003–2004 National Health and Nutrition Survey (NHANES) and physical activity data. They addressed missing data resulting from device failure in accelerometer measurement by implementing various imputation techniques known as additive regression, bootstrapping, and predictive mean matching (ARBP). The researchers carefully selected the most precise ARBP approach, which was then analyzed as the final model [12]. Additionally, in 2017, Beaulieu Jones and Moore [38] explored electronic health records (EHRs) as a valuable source of patient status data, despite the presence of extensive missing data.

As previously mentioned, stress has a significant impact on human health. Emotional pain, muscle aches, tension, digestive tract problems, and hyper arousal are all potential signs of acute stress. Some of the minor side effects of physical situations may including heartburn, back pain, stomach ache, headaches, rapid heartbeat and elevated blood pressure. However, it has a greater negative impact on human physical health. It is a significant risk factor for hypertension, irritable bowel syndrome and coronary disease [39], generalized anxiety disorder, gastro oesophageal reflux disease [40], and depression (http://www.bmj.com/content/315/7107/530, accessed on 30 August 1997)).

Psychological reactions involve the amplification of powerful negative emotions, such as anger, anxiety, irritation, or depression [41]. These responses can heighten our emotional experiences, leading to heightened feelings of worry, frustration, and hostility, thereby affecting our relationships [42]. From a physiological standpoint, increased activity in the sympathetic nervous system (SNS) alters the body’s hormone levels and triggers reactions such as increased sweating, elevated heart rate, and muscle activation [43]. Breathing becomes more rapid, and blood pressure rises [44]. These physiological changes also affect speech characteristics due to alterations in the muscles that regulate the respiratory system and vocal tract. Additionally, there is a decrease in skin temperature, as well as in the temperature of hands and feet [45]. The Heart Rate Variability (HRV) also decreases as a result [46].

The economy is significantly impacted by these health issues, leading to absenteeism, staff turnover, and tardiness among employees. Stress has also given rise to a widespread phenomenon known as “presenteeism”, where employees are physically present at their workstations but are not fully productive [32]. According to estimates, the cost of absenteeism and presenteeism in Europe is expected to be 269 billion euros annually, with an additional 239 billion euros in productivity losses [35]. Given the long-term consequences of stress, it is advisable to detect symptoms early on to prevent further damage.

The authors [47,48] conducted a comparison of various classification models, including Decision Tree (DT), Naïve Bayes (NB), K-Nearest Neighbors (KNN), Neural Network (NN), and Support Vector Machine (SVM). The findings of these studies revealed that the SVM classifier exhibited the highest accuracy. However, Kim et al. (2015) achieved superior accuracy (69.51%) than SVM by utilizing Fuzzy Logic and Decision Tree techniques on the sixth Korea National Health and Nutrition Examination Survey (KNHANES) dataset. On the other hand, [49,50] introduced DNN-based models for predicting Coronary Heart Disease (CHD). Atkov et al. (2012) constructed ten distinct prediction models using various risk factors. These models consisted of two hidden layers with four neurons each, and they achieved an accuracy of 93% using data from 487 patients at Central Clinical Hospital No. 2, of Russian. Furthermore, in [49,50], the authors performed feature correlation analysis and connected the hidden layers of the DNN based on the correlation results. The DNN model, integrated with feature correlation analysis, attained an accuracy of 83.9% and an AUC score of 79.0% on the sixth KNHANES dataset. 

## 3. Long Short-Term Memory 

A Long Short-Term Memory (LSTM) network is a special variant of conventional RNN, initially introduced by Sepp Hochraiter and Schmidhubar in 1997 [51]. LSTM networks are used to incorporate with an inception module; they have gained popularity due to their capability to examine the temporal relationship time-series data. Recently, the LSTM model reported state-of-the-art performance in various fields including text classification, speech recognition, machine translation and time series prediction. The capability of the LSTM network lies in its ability to effectively learn long sequential data and to propagate errors through all its layers. The standard LSTM was developed to address two major drawbacks commonly found in existing RNNs, namely gradient vanishing and expansion. LSTM included a gated mechanism and memory blocks that control the flow of information with self-connections.

The architecture of the conventional LSTM which contains memory blocks shown in Figure 2. As illustrated in Figure 2, the LSTM block provides a gating mechanism, which consists the three gates: the $i_{t}$, $o_{t}$, and $f_{t}$, which are the input, output, and forget gate of LSTM, respectively; $c_{t}$ refers to the memory cell state, ${\tilde{a}}_{t}$ is the candidate state computed by equation (4). $x_{t}$, $h_{t}$, and $h_{t-1}$ are the input, final output of the LSTM. An update to the cell state vector is calculated as in Equation (5). It is noted that these gate signals contain the logistic nonlinearity, presented by the Sigm activation function, and their signals range is between 0 and 1.

In order to determine the hidden state $\left(h_{t}\right)$ of an LSTM, a computation involving Equation (2) is performed on the output gate $o_{t}$. This computation is then multiplied by the cell state $c_{t}$ through the use of the *tanh* function in Equation (6). The weight matrix for the input is denoted by $W_{xo}$, while the weight matrix for the hidden state is represented by $U_{ho}$. Additionally, a bias term is included in the computation, which is indicated by $b_{0}$. Finally, the sigmoid function is also utilized *Sigm (x)* = $\frac{1}{1+e^{-x}}$. The most important part of the LSTM structure is ${\tilde{a}}_{t}$ and $c_{t}$. These functions retain the data for a long period. As a result, this mechanism supports the long-term dependencies. The mathematical equations of the LSTM memory block are summarized as follows: (1)$$i_{t}=Sigm\left(W_{xi}x_{t}+U_{hi}h_{t-1}+b_{i}\right)$$
(2)$$o_{t}=Sigm\left(W_{xo}x_{t}+U_{ho}h_{t-1}+b_{o}\right)$$
(3)$$f_{t}=Sigm\left(W_{xf}x_{t}+U_{hf}h_{t-1}+b_{f}\right)$$
(4)$${\tilde{a}}_{t}=tanh\left(W_{xȃ}x_{t}+U_{hȃ}h_{t-1}+b_{ȃ}\right)$$
(5)$$c_{t}=f_{t}*x_{t-1}+i_{t}*\;{\tilde{a}}_{t}$$
(6)$$h_{t}=O_{t}*tanh\;\left(c_{t}\right)$$

All parameters of the LSTM network are presented in Equations (1)–(6); in terms of weights and bias executed during the learning procedure, those processes involved are *Wi*, *Wo*, *Wf*, *Wȃ* ∈ *Rm × p*, *Ui*, *Uo*, *Uf*, *Uȃ* ∈ *Rm × m*, *bi*, *bo*, *bf*, and *bȃ* ∈ *R m* × 1. ∗ denotes point-wise multiplication of two vectors. Here ‘*tanh*’ is an element-wise hyperbolic tangent activation function.

## 4. Materials and Methods

In this study, we demonstrated the proposed framework which included enhanced LSTM with the combination of feature attention mechanism. The research design of this study is demonstrated in Figure 3. From the KNHANES VI dataset, this study considered various stress-related features from physical activity and lifestyle data, including systolic blood pressure, sleep time, body mass index, smoking, and drinking. The variables that were considered important for stress evaluation were extracted using statistical analysis of the selected data. 

The major contribution of the developed approach is the extraction of the most significant features into two key stages, such as pre-feature attention E-LSTM and post-feature attention E-LSTM. As a feature, the statistical analysis data was incorporated into the E-LSTM modeling. Furthermore, to evaluate the capability of the proposed E-LSTM approach, we compared the stress classification outcomes, achieved by employing the statistical analysis data with state-of-the-art techniques and the conventional LSTM model. 

## 5. Enhanced LSTM

LSTM networks are an advanced variant of traditional RNN, which was first developed by German authors Sepp Hochraiter et al., in 1997 [51], which are also followed by the existing architecture of RNNs with minor modifications [52]. We have already described the conventional LSTM network with a gating mechanism, which is depicted in Figure 2. However, here we introduced the enhanced LSTM architecture with modified equations that support the development of learning capabilities. The computational process of enhanced LSTM is not the same as the standard ones. In our enhancement, we used the “peephole connections mechanism” which is one of the most well-known variations of the LSTM architecture. Figure 4 presents the enhanced architecture of LSTM with peepholes.

In this section, we applied an enhanced LSTM architecture, which allows the gate layers to look at the cell state and adapt the peephole connection mechanism that completely influences the gating mechanism referred as the following equations:(7)$$i_{t}=\varphi \left(W_{xi}\times \left[\;C_{t-1},h_{t-1},x_{t}\right]+b_{i}\right)$$
(8)$$f_{t}=\varphi \left(W_{xf}\times \left[\;C_{t-1},h_{t-1},x_{t}\right]+b_{f}\right)$$
(9)$${\bar{C}}_{t}=tanh\left(W_{x\bar{c}}*\left[h_{t-1},x_{t}\right]+b_{\bar{c}}\right)$$
(10)$$C_{t}=f_{t}*C_{t-1}+i_{t}*\;{\bar{C}}_{t}$$

In these equations, the input of transition matrices presented by $x_{t}$, the memory cell $C_{t-1}$, while ∗ referred the element-wise multiplication, $h_{t-1}$ denotes the hidden state vector and *φ* shows the non-linearity sigmoid activation function. The output gate $o_{t}$ controls the current hidden state value $h_{t}$ using the system nonlinearity to the contents of the memory cell:(11)$$o_{t}=\varphi \left(W_{xo}\times \left[C_{t},h_{t-1},x_{t}\right]+b_{o}\right)$$
(12)$$h_{t}=o_{t}*tanh\;\left(C_{t}\right)$$

In the next phases, the current time of the hidden state $h_{t}$ is employed to acquire $h_{t+1}$. Moreover, long short-term memory computes the internal hidden state $h_{t}$, of the word series sequentially at each time step. The final time step of hidden activations can be used as the input to the layer that classifies stress by providing a semantic representation of the whole sequence.

### 5.1. Sequential Mechanism by Pre-Feature Attention E-LSTM

In order to make full use of the unique sentiment resource information in the sentiment analysis tasks, this study utilizes both pre-feature attention and post-feature attention mechanisms incorporating with the E-LSTM model. Our methodology performs well on computing feature-level and sentence-level attention, which is employed to integrate information from both word and sentence proceeding through the pre-feature attention mechanism for stress identification. Usually, due to the longer length of the input series, it is a challenging task for the conventional LSTM model to extract significant features during the training process for stress prediction [53]. 

However, specific features perform an important role and contribute to building a model that can accurately classify data. Similarly, our feature attention method is a vital component of the improved E-LSTM architecture, which helps extract the most informative features from challenging stress-related reviews to classify emotions based on word-level presentation [54]. In addition, the LSTM utilizes a gating strategy to manage information flow within memory, while the two-state LSTM structure combines information from both preceding words within desirable contexts [55]. Furthermore, for accurate stress identification, the forward and backward sub-states are included in the pre-feature attention mechanism modeling. 

The sub-states in the neural network are responsible for processing the input data. The forward sub-state captures the words in sequence from the start of the input layer to the end. Conversely, the backward sub-state performs computations in the reverse order, as the forward sub-state performs computations in the forward direction. Typically, during a time step t, the input feature $x^{k}$, is used to initialize the forward and backward candidate states ${\vec{h}}_{t-1}$ and ${\overset{\leftarrow }{h}}_{t-1}$, which are then used in the pre-feature attention E-LSTM. The previous and current states of the forward ${\vec{C}}_{t}$ and backward ${\overset{\leftarrow }{C}}_{t}$ sub-states are also considered in this process and are referred to as follows: (13)$${\bar{C}}_{t}=tanh\left(\vec{W_{x}^{\left(\bar{C}\right)}}*[\vec{h_{t-1},x_{t}]}+\vec{b_{\left(\bar{C}\right)}}\right)$$
(14)$${\bar{C}}_{t}=tanh\left(\overset{\leftarrow }{W_{x}^{\left(\bar{C}\right)}}*\overset{\leftarrow }{\left[h_{t-1},x_{t}\right]}+\overset{\leftarrow }{b_{\left(\bar{C}\right)}}\right)$$
(15)$$\bar{C}=ƒ\left(V\left[\vec{{\bar{C}}_{t}}\;:\;\overset{\leftarrow }{{\bar{C}}_{t}}\right]+k\right)$$

### 5.2. Feature Seizing by Attention Mechanism

After producing the final output by hidden state, we incorporated an attention mechanism into the feature-attention process to assist the architecture in identifying stress detection by focusing on valuable information at the word-feature level. Our proposed mechanism utilizes a detailed attention structure, which is depicted in Figure 5, while Figure 5 illustrates the distribution of attention generated at a specific time step $o_{t}^{k}$ at $k^{th}$ by the attention mechanism as follows:(16)$$o_{t}^{k}=\frac{exp\left(e_{t}^{k}\right)}{\sum_{i=1}^{m}exp\left(e_{t}^{i}\right)}$$
where the score function for memory cell $c_{t-1}$ at $k^{th}$ at a specific time step $e_{t}^{k}$ is denoted by $k^{th}$, and score function demonstrated by $e_{t}^{k}$:(17)$$e_{t}^{k}=\left[c_{t-1}^{T}h_{1},\;c_{t-1}^{T}h_{2},\ldots ,c_{t-1}^{T}h_{m}\right]$$
where the pre-feature attention E-LSTM utilizes hk to represent its hidden unit. Subsequently, the score functions derived from it are employed by the post-feature attention LSTM. The resulting attention output is then obtained through this process as follows:(18)$$o_{t}=\sum_{k=1}^{m}o_{t}^{k}h_{k}$$

### 5.3. Post-Feature Attention LSTM

In this mechanism, the proposed approach utilizes a feature attention mechanism to learn stress level information that is intentionally similar to human behavior. This is achieved through a post-feature attention E-LSTM, which is followed by a word-feature seizing mechanism. In the second phase of this approach, a post-feature attention E-LSTM is used to imitate the decoded function. This step involves extracting predicted features that have been generated by both the pre-feature attention E-LSTM and the attention mechanism layer. The major equation used in the post-feature attention E-LSTM is equivalent to that of a standard LSTM, as follows: (19)$${\bar{C}}_{t}=tanh\left(W_{x}^{\bar{C}}*\left[h_{t-1},x_{t}\right]+b_{\bar{C}}\right)$$

The process of predicting stress levels in the KNHANES VI dataset involves transforming the output features vector of post-feature attention E-LSTM into a sentence representation through a dense layer, followed by the use of a sigmoid activation function for final classification into “Low stress” or “High stress” categories. The accuracy of the model is greatly impacted by the extraction and selection of features, which directly affects its performance. As a result, we developed a feature attention enhanced LSTM mechanism specifically for stress classification. The detailed architecture of this mechanism is depicted in Figure 6.

## 6. Experimental Design

This research simulations were conducted on a computer with an Intel “core-i7-3770CPU @ 3.40 GHz” processor and 8 GB of RAM, running on the Windows 10 operating system. Python 3.9 compiler and Anaconda were utilized as the development environment for data pre-processing and analysis, along with TensorFlow 1.14 and Keras 2.3 as the necessary libraries. Additionally, to optimize the developed mechanism, a brief description of the datasets and implementation hyperparameter settings were provided in the following subsections. 

### 6.1. Dataset

This research examined the records of adults aged 18–75 from the dataset attained by the Health Questionnaire and Nutrition Survey conducted during the 2013–2015 KNHANES VI. In KNHANES VI (2013–2015), a set of domestic experiments were carried out to establish the correlation between physical activity, lifestyle, and stress. The survey responses were categorized into four stress groups to measure the stress levels of the participants. To determine stress classification, a set of variables, including age, gender, sleeping duration, pulse rate, body mass index (BMI), systolic and diastolic blood pressure (SBP and DBP), height, weight, smoking and drinking habits were utilized as input parameters for learning, while the output variables indicated whether the subjects were stressed or not. There was a total of 22,898 experimental records from KNHANES VI (2013–2015)”. There were 14,622 record totals, except the unknown (non-respondent, null value) respondents. There were 651 people who felt extremely stressed out of 14,619 records, and 2529 people who did not feel stressed. As a result, the insufficient number of people experiencing stress led to the labeling of two groups: the group with low levels of stress was categorized as such because they did not feel stressed, while the group with high levels of stress was labeled as such because they felt significantly stressed.

### 6.2. Deep Learning Environment

In this study, The E-LSTM model was developed using a Deep Learning (DL) environment based on the TensorFlow platform and Python for designing, training, and distributing neural networks. Hyperparameters within the DL platform are variables that play a crucial role in determining how a neural network learns through the use of Python syntax. The configuration of visible and hidden layers, along with their corresponding activation functions (such as tanh and sigmoid), can be declared and constructed using the multilayer-configuration object to optimize the learning process. Generally, the algorithmic complexity of an algorithm is typically expressed as O(W), where W represents the estimated number of parameters in the network. The calculation of W commonly involves to considering two variables: the dimension of the input vector (**m-dimension**) and the dimension of the hidden layer (**n-dimension**). Table 1 provides the estimated parameter counts for various deep learning approaches, including NB, SVM, LSTM, DBN, Bi-LSTM, and our proposed E-LSTM.

In comparison to the traditional LSTM, our proposed E-LSTM has a higher inner complexity due to the involvement of a larger number of parameters. Consequently, the execution of E-LSTM requires more time and resources compared to conventional NB, SVM, DBN, and LSTM. However, when compared to the existing approach of Bi-LSTM, our proposed mechanism is less complex, resulting in reduced execution time. Importantly, our proposed feature attention mechanism has the capability to extract valuable information, leading to a significant enhancement in stress classification accuracy.

### 6.3. Hyperparameter Setting 

In a deep learning study, the effectiveness of a deep learning model is typically dependent on various factors, including the quantity of hidden layers and nodes, as well as the selection of appropriate hyperparameters. In this study, we experimented with the number of layers, nodes and hyperparameters to develop an effective E-LSTM based model for stress classification. Our experiment utilized the Adam optimizer with default optimal parameter settings, including a learning rate of 0.0005 and a decay factor of 0.9. The application of deep learning-based methods allows for the acquisition of an understanding of intricate connections between inputs and outputs [56]. The term ‘learning rate’ refers to the amount of time it takes to change the value of a parameter, whereas ‘momentum’ refers to the ability to accelerate or decelerate parameter. 

Additionally, in each training iteration we maintain a fixed batch size of 768, which presents the grouping of multiple input data. To mitigate the risk of overfitting, we have incorporated the dropout technique [57], with a dropout rate of 0.2 for the LSTM layer and $L_{2}$ regularization with $10^{-5}$ for the coefficient λr. ‘$L_{2}$ regularization’ is a usually applied for normalization method to avoid the overfitting. Likewise, in this study, we employed binary cross-entropy as the loss function along with $L_{2}$ regularization, which is defined as follows:(20)$$J\;\left(w,\;b\right)=-\frac{1}{N}\sum_{i=1}^{N}[y_{i}\cdot log{\hat{y}}_{i}+\left(1-y_{i}\right)\cdot log\left(1-{\hat{y}}_{i}\right)]$$
where $y_{i}$ is the refer label; and classification probability is represented by ${\hat{y}}_{i}$. We set w = 0.001, of Fresenius norm value by compressing $L_{2}$, which is the coefficient for $L_{2}$. In the training phase, it was observed that utilizing $L_{2}$ regularization and dropout techniques can enhance performance by preventing overfitting. The optimal values of hyperparameters that were utilized for training the proposed framework are presented in Table 2.

### 6.4. Evaluation Metrics 

Various assessment measures have been employed to evaluate the efficacy of the developed model in predicting stress-related issues when compared to other approaches. These evaluation metrics included: accuracy, precision, recall, and F-measure.
(21)$$Accuracy=\frac{\left(TP+TN\right)}{\left(TP+TN+FP+FN\right)}$$
(22)$$Precision=\frac{TP}{\left(TP+FP\;\right)}$$
(23)$$Sensitivity=Recall=\frac{TP}{\left(TP+FN\;\right)}$$
(24)$$F1-Score=2\times \frac{Precision\times Recall}{Precision+Recall}$$
where TP, TN, FP and FN are the True Positive, True Negative, False Positive and False Negative, respectively.

## 7. Results and Discussion

The experimental results provide a concise summary of the simulation outcomes obtained using the proposed E-LSTM approach, which is then compared to state-of-the-art approaches using “KNHANES VI (2013–2015) datasets. The results are then examined based on different evaluation metrics. 

### 7.1. Statistical Analysis

Table 3 referred to the distribution of physical and lifestyle activities based on low- and high-stress categories for 1280 stress records. In this study, two kinds of tests were performed. A comparison was made between specific average age, sleep duration, pulse rate, DBP, SBP, weight, height, and BMI between high-stress and low-stress classes; a t-test was also conducted. Meanwhile, a Chi-square test was used to investigate the relationship among gender, smoking, drinking variables, and stress. Both the t-test and Chi-square test used a significance level of *p* < 0.05 to determine the suitable variables for classifying stress. There were nine variables identified as significantly associated with stress based on their *p*-value less than 0.05. These variables were gender, age, sleep duration, pulse rate, SBP, weight, height, drinking,” and smoking. 

The classification ability was assessed using a confusion matrix, which served as a performance evaluation tool for the model. Specifically, in terms of accuracy, precision, sensitivity, and F1-score were calculated as depicted in Figure 7, while confusion matrix results are illustrated in Table 4. The matrix consisted of the classification results (low stress, high stress) of the test dataset related to stress levels.

Figure 8 presents the loss function graph of the proposed approach in the term of configuration; (A) configuration 1 and (B) configuration 2, for stress classification dataset. On the graph, the epoch is displayed on the x-axis while the loss is depicted on the y-axis. The loss function has been captured based on 100 and 150 epochs (the left side of Figure 8A contains 150 epochs and the right-side of Figure 8B contains 100 epochs). However, the proposed approach maintains their loss rate after 140 epochs in configuration 1, while in the configuration 2, the model maintains their loss rate between 13 and 37 iterations. The loss function is a mathematical expression that computes the difference between the real and predicted outputs. 

In the experiment, the two model configurations were compared by adjusting different parameters and scaling the graph of the loss function accordingly. Analyzing the results, we observed that the model configuration 1 has the minimum loss value. Therefore, in the terms of loss function and accuracy, we considered configuration 1 as the most favorable model.

### 7.2. Accuracy and Comparison

In this study, we used physical activity and lifestyle data to classify stress based on enhanced LSTM. We executed the implementation of our proposed E-LSTM model using KNHANES VI (2013–2015) datasets with particular parameters. Model training was carried out using stochastic gradient descent through considered batch size. 

In our work, we assessed the stress analysis performance of the proposed E-LSTM model with comparative approaches such as naïve Bayes (NB), support vector machine (SVM), deep belief network (DBN) and long short-term memory (LSTM). Our developed E-LSTM model achieved excellent performance in the term of accuracy. Figure 9 shows the evaluation results of the proposed model and state-of-the-art approaches using the KNHANES-VI dataset.

### 7.3. Model Performance in Terms of Precision, Recall and F1-Score

A thorough performance analysis was conducted on both the proposed and comparative models, which involved to evaluate their performance using various metrics such as precision, recall, and f-measure. Precision refers to the number of true positive observations that are correctly classified as complete positive instances. Recall measures the ability to accurately identify positive instances among all real class instances. To obtain the final values of precision and recall, we separately calculated these steps and take the ratio of all classes. Finally, f-measure was used to determine the test accuracy, which was computed using the test precision and recall.

In this section, Figure 10 and Figure 11 present the performance of the proposed E-LSTM model and comparison with exiting approaches using KNHANES VI dataset. These figures show that the proposed approach achieved higher classification accuracy than the existing approaches, due to the efficient pre-processing and effective classification using the feature attention mechanism.

Figure 12 illustrates the comparison results of the proposed model with state-of-the-art approaches in the terms of accuracy, precision, recall and F1-score. The proposed E-LSTM model obtained an accuracy of 75.54%, precision of 74.26%, recall of 72.99%, and F1-score of 74.58%. Conversely, the E-LSTM model has the advantage of saving time as it allows for the use of unlabeled training samples due to its supervised learning mechanism. As a result, the E-LSTM model performs better in terms of human labelling time.

### 7.4. Discussion

In this study, we explored the classification of daily life stress using Enhanced LSTM (E-LSTM) based on the feature attention mechanism that focuses on determining and categorizing the stress polarity using sequential modelling and word-feature seizing. The goal was to develop a model that could accurately predict the level of stress individuals experience in their day-to-day lives. The results of our experiments demonstrate the potential of E-LSTM models for stress classification and offer valuable insights into the factors influencing stress levels.

Our findings indicate that the E-LSTM approach can effectively capture the temporal dependencies present in daily life stress data. By considering the sequential nature of stress experiences, E-LSTM networks can learn patterns and relationships across time, which are crucial for accurate stress classification. This is particularly relevant in the context of stress, as stressors often occur in succession and can have cumulative effects. Furthermore, we investigated the useful functionality of the conventional LSTM for stress classification using stress-correlated physical activity and lifestyle data accessed from the 2013–2015 Korea National Health and Nutrition Examination Survey (KNHANES VI) database [31]. First, we studied the stress assessment was feasible by analyzing the stress related physical activity and lifestyle data and divide the persons between the ages of 18 to 75 years old into two groups: those who were frequently felt stressed and those who did not. This dataset (KNHANES-VI) consists of 14,620 experimental records. By analyzing the statistical data, we confirmed that our proposed E-LSTM model performed successfully on stress classification data.

We compared the performance of our developed enhanced (E-LSTM) approach with several baseline approaches, including naïve Bayes, support vector machine, long short-term memory, deep brief network and bidirectional LSTM. Our experimental results demonstrate that the proposed (E-LSTM) approach outperformed the baseline models in terms of stress classification accuracy. The LSTM model achieved accuracy, precision, recall and F1-scores of 75.54%, 74.26%, 72.99% and 74.58%, respectively. This improvement in accuracy highlights the efficacy of proposed approach for stress classification and their ability to capture the complex dynamics of stress experiences.

Moreover, in our proposed study, we explored some pros and cons of using the enhanced LSTM (E-LSTM) approach for daily life stress classification. Our findings shed light on the potential benefits and limitations of employing E-LSTM model. It is worth noting that our study has a few limitations. Firstly, in comparison to the traditional LSTM, our proposed E-LSTM has a higher inner complexity due to the involvement of a larger number of parameters. Consequently, the execution of E-LSTM requires more time and resources compared to conventional NB, SVM, DBN, and LSTM. However, when compared to the existing approach of Bi-LSTM, our proposed mechanism is less complex, resulting in reduced execution time. Importantly, our proposed feature attention mechanism has the capability to extract valuable information, leading to a significant enhancement in stress classification accuracy. Secondly, the stress classification model used in this study was unable to determine the degree of stress in the two subcategories of stress. Therefore, a more in-depth analysis of the degree of stress is required to design a more reliable stress classification system that can thoroughly examine the degree of stress.

In conclusion, our study demonstrates the effectiveness of E-LSTM approach for use in stress monitoring systems for stress prediction. The results highlight the ability of the E-LSTM model to capture temporal dependencies and learn patterns in stress data, leading to improved stress classification accuracy. The insights gained from feature importance analysis provide a better understanding of the factors influencing stress levels. Regarding the direction of future study, future research should focus on addressing the limitations of this study and further exploring the potential of E-LSTM models for stress classification in various real-world applications.

## 8. Conclusions and Future Scope

This paper has presented an effective stress classification model using a deep learning method by integrating the Enhanced LSTM (E-LSTM) model through a feature-attention mechanism. Our study involved the development of a new feature-attention mechanism that utilizes both pre- and post-feature attention layers to obtain more comprehensive feature representation. By implementing this mechanism, we were able to construct an efficient stress classification model that was evaluated using stress-related health data obtained from the KNHANES VI (2013–2015) dataset, consisting of 14,620 experimental records. By analyzing the statistical data, we confirmed that our proposed E-LSTM model performed successfully on stress classification data. The experimental results showed that the proposed feature-attention E-LSTM framework achieved excellent results with an accuracy of 75.54%, precision of 74.26%, and recall of 72.99%, respectively. The proposed E-LSTM model performed better than other state-of-the-art classification approaches, namely, naïve Bayes, SVM, deep belief network and LSTM. This research demonstrated the effectiveness of the proposed method in identifying stress detection, indicating its potential for use in stress monitoring systems for stress prediction.

However, there are some limitations that need to be acknowledged in this study. Specifically, the stress classification model used in this study was unable to determine the degree of stress in the two subcategories of stress. Therefore, a more in-depth analysis of the degree of stress is required to design a more reliable stress classification system that can thoroughly examine the degree of stress.

## Figures and Tables

**Figure 1 brainsci-13-00994-f001:**
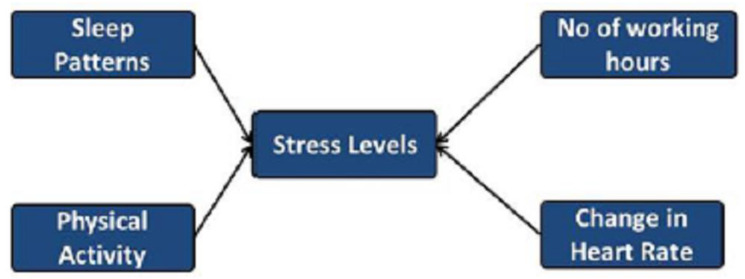
The conceptual design of stress levels.

**Figure 2 brainsci-13-00994-f002:**
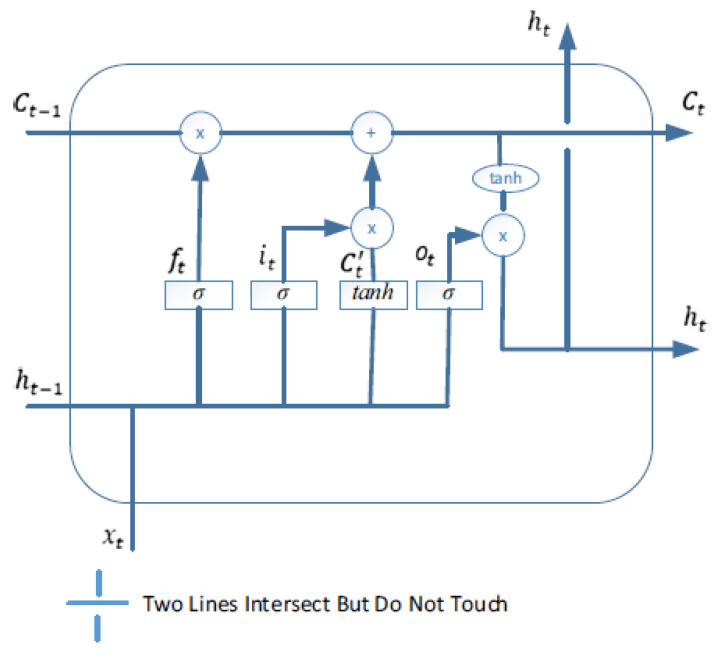
The LSTM architecture.

**Figure 3 brainsci-13-00994-f003:**
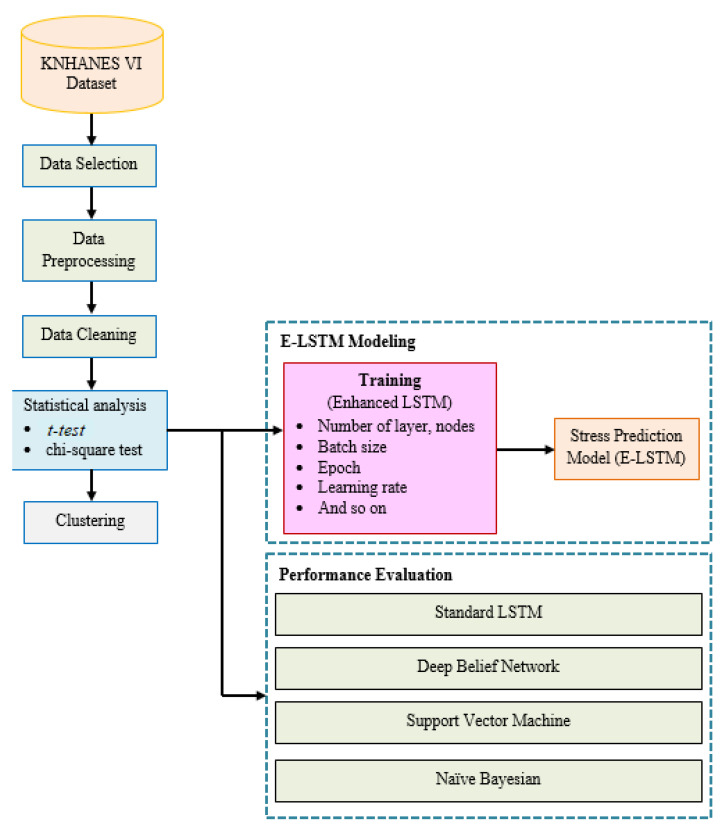
The overall framework of proposed study.

**Figure 4 brainsci-13-00994-f004:**
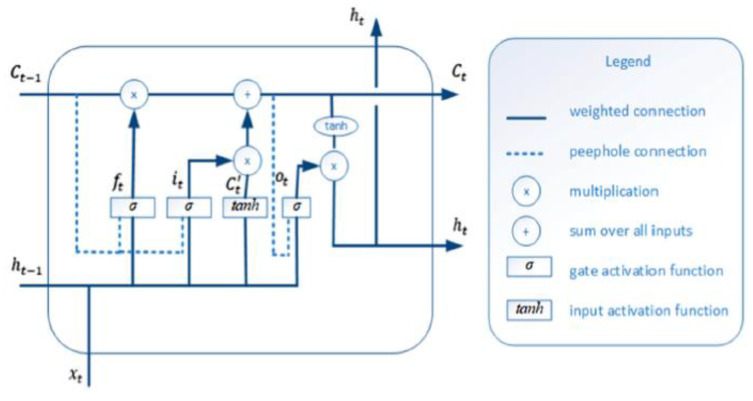
Enhanced LSTM architecture with peepholes.

**Figure 5 brainsci-13-00994-f005:**
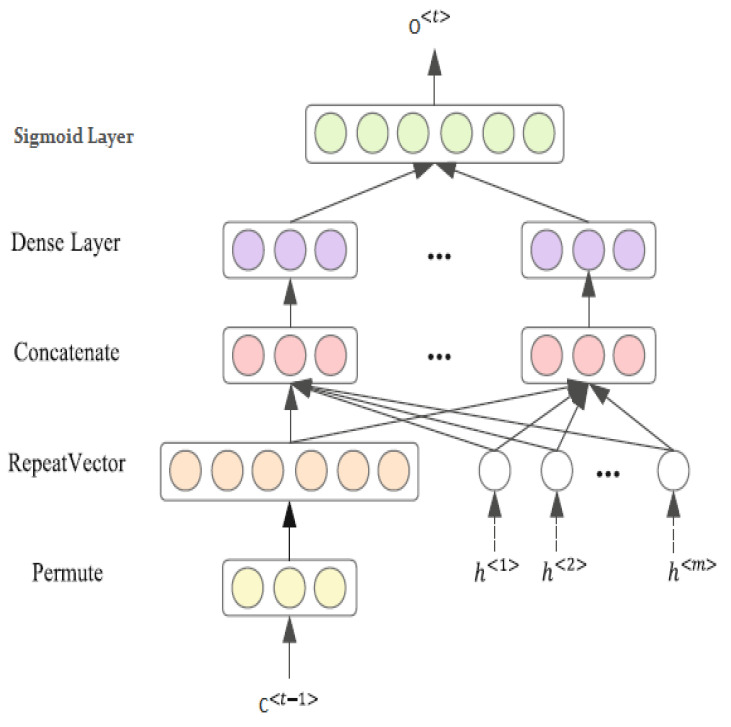
The detailed structure of the attention mechanism.

**Figure 6 brainsci-13-00994-f006:**
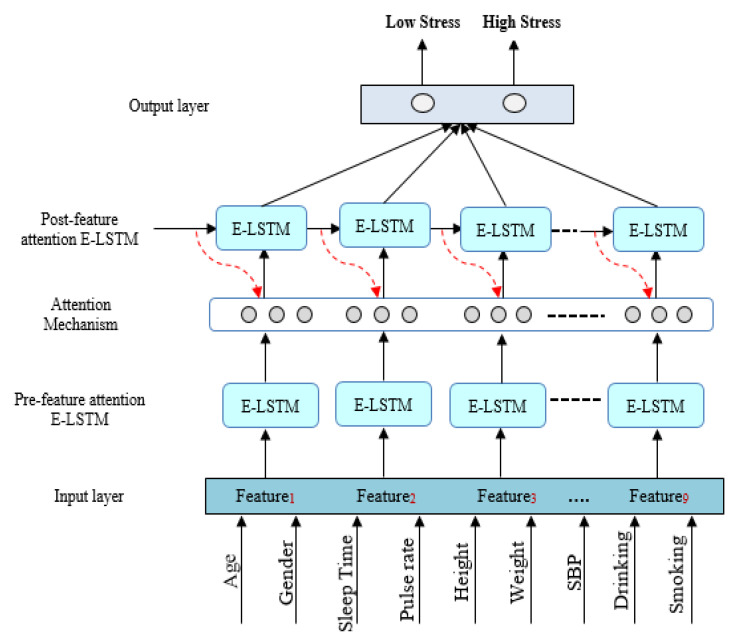
The overall architecture of proposed feature-attention E-LSTM.

**Figure 7 brainsci-13-00994-f007:**
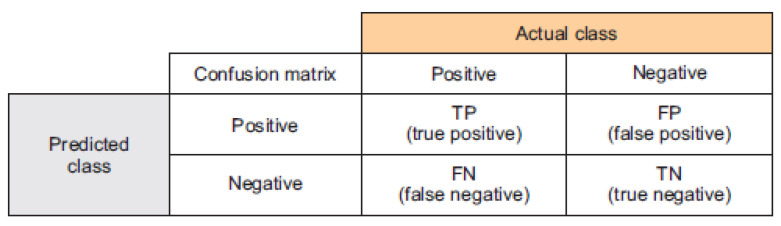
Confusion matrix.

**Figure 8 brainsci-13-00994-f008:**
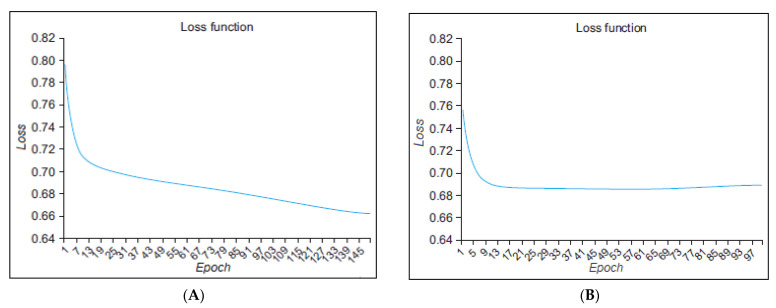
Loss function of proposed approach in the terms of configuration: (**A**) configuration 1, (**B**) configuration 2.

**Figure 9 brainsci-13-00994-f009:**
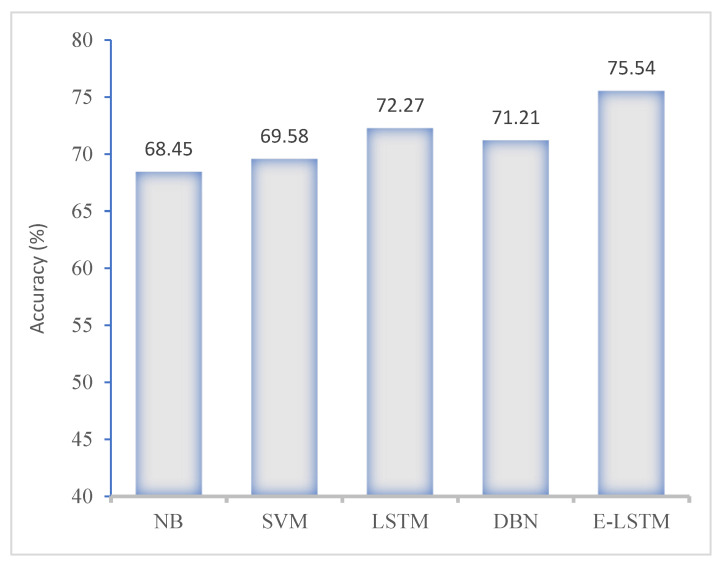
Accuracy comparison of proposed model with existing approaches.

**Figure 10 brainsci-13-00994-f010:**
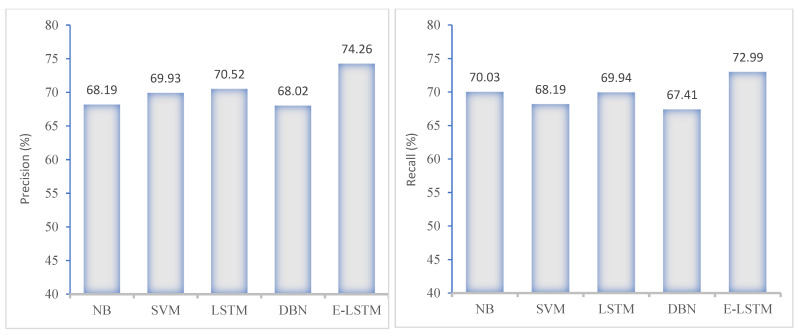
Precision and recall comparison of the proposed model with the existing approaches.

**Figure 11 brainsci-13-00994-f011:**
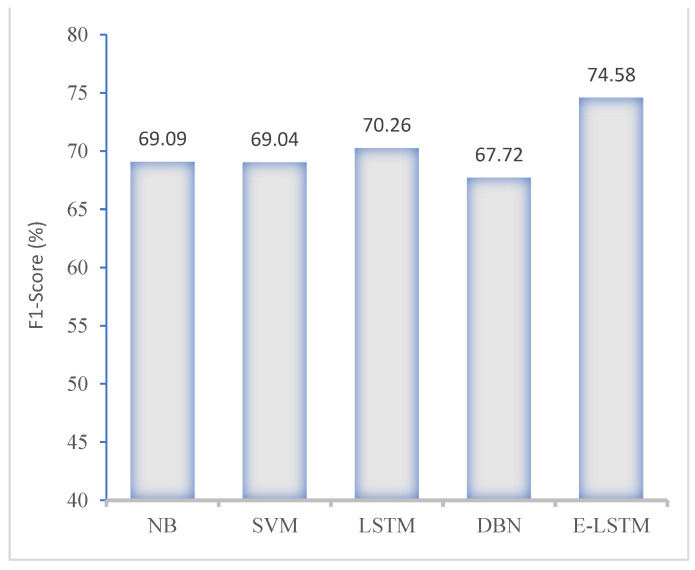
F1-score comparison of proposed model with existing approaches.

**Figure 12 brainsci-13-00994-f012:**
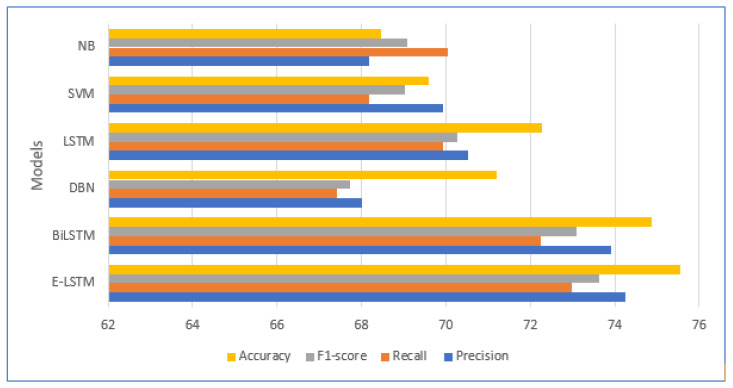
Comparison performance of the model with state-of-the-art approaches.

**Table 1 brainsci-13-00994-t001:** Numbers of parameters of proposed and traditional approaches.

Approaches	Numbers of Parameters
NB	$2\times \left(n^{2}+nm+n\right)$
SVM	$2\times \left(n^{2}+nm+n\right)$
GRU	$3\times \left(n^{2}+nm+n\right)$
LSTM	$4\times \left(n^{2}+nm+n\right)$
DBN	$6\times \left(n^{2}+nm+n\right)$
BiLSTM	$8\times \left(n^{2}+nm+n\right)$
**Proposed E-LSTM**	$7\times \left(n^{2}+nm+n\right)$

**Table 2 brainsci-13-00994-t002:** Optimal hyperparameters of the proposed E-LSTM model.

Hyperparameters	E-LSTM	
	Model Configuration 1	Model Configuration 2
Activation Function	Binary Cross-entropy	Sigmoid
Number of input nodes	9	9
Number of output nodes	2	2
Learning Rate	0.0005	0.0003
Batch Size	768	1024
Dropout Rate	0.2	0.3
Epoch	150	100
Momentum	0.1	0
Regularizer	L2	L2
Accuracy (%)	75.54	73.98

**Table 3 brainsci-13-00994-t003:** Distribution of physical activity and lifestyle between low and high-stress.

Physical Activity and Lifestyles	Low Stress	High Stress	p-Value
Gender			0.001
Man	289	228	
Woman	351	405	
Avg. age (year)	56.72	47.54	0.000
Avg. height (cm)	159.80	161.18	0.014
Avg. weight (kg)	63.12	64.07	0.049
Avg. sleep duration (hr)	6.50	6.15	0.000
Avg. pulse rate (bpm)	70.10	72.68	0.000
Avg. SBP (mmHg)	120.92	116.37	0.000
Avg. DBP (mmHg)	73.52	74.59	0.190
Avg. BMI (kg/m^2^)	24.08	24.16	0.670
Drinking			0.004
No	338	287	
Yes	290	341	
Smoking			0.000
No	542	474	
Yes	98	162	

DBP: diastolic blood pressure, SBP: systolic blood pressure, and BMI: body mass index.

**Table 4 brainsci-13-00994-t004:** Confusion matrix results.

Models	True Positive	False Positive	False Negative	True Negative
NB	712	98	182	242
SVM	894	14	369	8
LSTM	902	20	348	16
DBN	840	46	178	86
BiLSTM	910	88	158	202
Proposed E-LSTM	926	76	142	212

## Data Availability

The data is publicly available at Kaggle: https://www.kaggle.com/datasets/cdc/national-health-and-nutrition-examination-survey (accessed on 12 June 2022).

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
