# Peer review of "Attention Aware Deep Learning Approaches for an Efficient Stress Classification Model"

_brainsci, 2023, doi:10.3390/brainsci13070994_

Round 1

Reviewer 1 Report

The main objective of the research study was to find an efficient model for stress classification which is interesting and relevant to today’s scenario. The study findings support the framed objective. Though the topic is not new as there are other similar models, the aim of the study is appreciated as an existing model was modified to achieve a better outcome suited to the current societal needs. The entire text has to be revised as extensive grammar corrections have to be made. The paper needs to be rewritten with more emphasis on discussion adding pros and cons of other existing models and explanation has to be given in depth supporting the findings of the current study model as already there are other existing models.

Extensive english correction needs to be done throughout the article

Reviewer 2 Report

This manuscript presents a novel stress classification method based on LSTM. The authors quantitatively evaluated its performance using a large-scale dataset collected by a reputable public institution. The comparative experimental results demonstrate that the proposed method, named E-LSTM, outperforms Vanilla LSTM. These results indicate the novelty and superiority of the proposed method in terms of accuracy. Therefore, the proposed method and experimental results are highly commendable, indicating the potential value of this manuscript as an academic paper. Nevertheless, the current version of the manuscript has several unclear points and unresolved questions that need to be addressed.

1. The choice of the model for learning and predicting time-series features is not suitable as a comparative method. This is primarily due to the outdated nature of NB, SVM, and DBM approaches. In addition to Vanilla LSTM, a comprehensive exploration of LSTM derivative models, such as BLSTM (bidirectional LSTM; https://doi.org/10.1016/j.neunet.2005.06.042), should be undertaken to establish appropriate comparative methods. The results should demonstrate the superiority of E-LSTM in comparison to state-of-the-art classifiers.

2. The survey and investigation of related studies, including the comparative methods, are inadequate considering the diverse range of research examples already present in this research field. However, the introduction in Chapter 1 presents very limited coverage of related studies and their analysis. On the other hand, Chapter 2 discusses background information under the title "Stress Impact on Society and Economy." This content should be summarized in Chapter 1. Furthermore, Chapter 2 should allocate sufficient space to comprehensively describe the related studies in this specific research field.

3. While E-LSTM demonstrated slightly higher accuracy compared to Vanilla LSTM, the manuscript lacks a comparison of memory capacity and computational speed, which results in an unclear understanding of the trade-off relationship between these factors. When employing machine learning algorithms, it is essential to validate from multiple perspectives, including parameter dependency and ablation experiments. Which are the hyperparameters? Are there any limitations or drawbacks of your method?

4. Figure 8 lacks units on its axes, and Figures 9 to 11 can be represented in a single bar graph with a legend. Figure 7 does not provide any value as a figure in an academic paper since this is not a textbook. The manuscript fails to adequately present the experimental conditions and results due to the inadequacy of these drawings.

5. Was the evaluation conducted using cross-validation? If not, could you please provide an explanation as to why these drawings are inadequate in presenting the experimental conditions and results? If it was performed, the results should be presented. Specifically, Figures 9 to 11 should include vertical bars indicating the variance to accurately represent the range of data.

6. The learning curve in Figure 8 should illustrate the trends for both training and validation sub-datasets to provide a comprehensive view of the learning process

7. Stress stages should be classified into more than just "high" and "low." Why are only two categories used? In cases where there is no stress at all, why is it still classified as "low"?

8. Stress levels vary among individuals. How does E-LSTM learn individual differences? The analysis of misclassification results, which account for about a quarter of the cases, is not provided, making the causes of errors unclear.

Round 2

Reviewer 1 Report

Brain Sciences Manuscript ID: brainsci-2447699 –COMMENTS FOR REVISED REVIEW

Extensive revision has been made as per the comments given. Still the following issues need to be addressed:

1.     All relevant “pro and con” references justifying the given have been added in the “Introduction” section. This has to be shifted to the “Discussion” section. A separate discussion section is a must and has to be created and added as the current one has the heading “Results and Discussion” where only the results are described.

2.     The entire manuscript still requires English grammar editing throughout. An efficient English editing service can be utilized for the same and be remodified.

Reviewer 2 Report

The revised manuscript has been thoroughly improved and revised based on the previous feedback. As a constructive suggestion, I recommend addressing the following two aspects to further enhance the quality of your manuscript:

1. Please consider incorporating labels and units on the x-axis of Figures 8 and 12.

2. Since Figure 12 has been added, it may be prudent to reassess the necessity of Figures 9-11.
